# CellDuality: Unlocking Biological Reasoning in LLMs with Self-Supervised RLVR

**Yuhang Chen**[1]   **Zhen Tan**[2]   **Ruichen Zhang**[1]   **Mufan Qiu**[1]   **Tianlong Chen**[1]
[1]University of North Carolina at Chapel Hill   [2]Arizona State University
{yuhang, ruichen, mufan, tianlong}@cs.unc.edu  ztan36@asu.edu

## Abstract

Developing generalist large language models (LLMs) capable of complex biological reasoning is a central challenge in computational biology. While existing LLMs excel at predictive tasks like cell type annotation and logically-constrained problems, enabling open-ended and mechanistic reasoning remains a challenge. A promising direction is Reinforcement Learning from Verifiable Rewards (RLVR), which has been shown to significantly enhance complex reasoning in general domains like mathematics and code synthesis. However, its application in biology is hindered, as most biological outcomes are non-verifiable. For example, verifying a generated gene sequence is usually infeasible. In this paper, we introduce CellDuality, a self-supervised framework that enables LLM agents for robust reasoning in single-cell biology. Our framework is built on the principle of complementary task duality, a self-verification process that leverages a bidirectional reasoning loop. First, the model performs a forward reasoning task by predicting a biological outcome (e.g., a cell's response to a drug). Then, in a complementary inverse task, it must reason backward from its own prediction to reconstruct the initial conditions (e.g., the original drug perturbation). The fidelity of this reconstruction serves as an intrinsic reward signal, creating a feedback loop that enforces logical and biological consistency. We use these intrinsic rewards to align the base LLM via reinforcement learning, without requiring ground-truth verification labels. We demonstrate that CellDuality achieves state-of-the-art performance and provides coherent biological explanations across a diverse suite of single-cell reasoning tasks. Critically, on the challenging out-of-distribution perturbation prediction benchmark, our self-supervised approach significantly outperforms the standard fine-tuning baseline and narrows the performance gap to a supervised RLVR baseline. Our work showcases a new path toward scalable training of biological foundation models.

## 1 Introduction

Developing generalist large language models (LLMs) capable of biological reasoning is a central goal of computational biology (Fang et al., 2025b; Istrate et al., 2025; Lotfollahi et al., 2019). This reasoning ability involves inferring complex, mechanistic principles from cellular data (Fang et al., 2025b; Matsumoto et al., 2025). This capability is paramount in single-cell biology, where understanding causal chains, such as how a cell responds to a drug, is key to therapeutic discovery (Fang et al., 2025a). However, achieving robust biological reasoning is fundamentally challenging due to the stochastic nature of cellular systems and the intricate, high-dimensional dependencies between biological entities. This complexity creates a significant hurdle for current methods, especially the foundation models (Hao et al., 2024; Cui et al., 2024), which we categorize into three limitations.

First, most models are optimized for *prediction*, not *mechanistic reasoning*. Architectures like scGPT (Cui et al., 2024) and C2S-Scale (Rizvi et al., 2025) excel at learning correlational patterns for tasks like cell type annotation but are not explicitly trained to generate the coherent, explanatory steps that capture underlying biological pathways. Second, existing reasoning-aware models often operate in *logically-constrained paradigms*. For instance, Cell-o1 (Fang et al., 2025b) models a deductive puzzle-solving process rather than the open-ended, hypothesis-driven inquiry of scientific exploration. Finally, there exists a trade-off between *depth and generality*. Specialized models

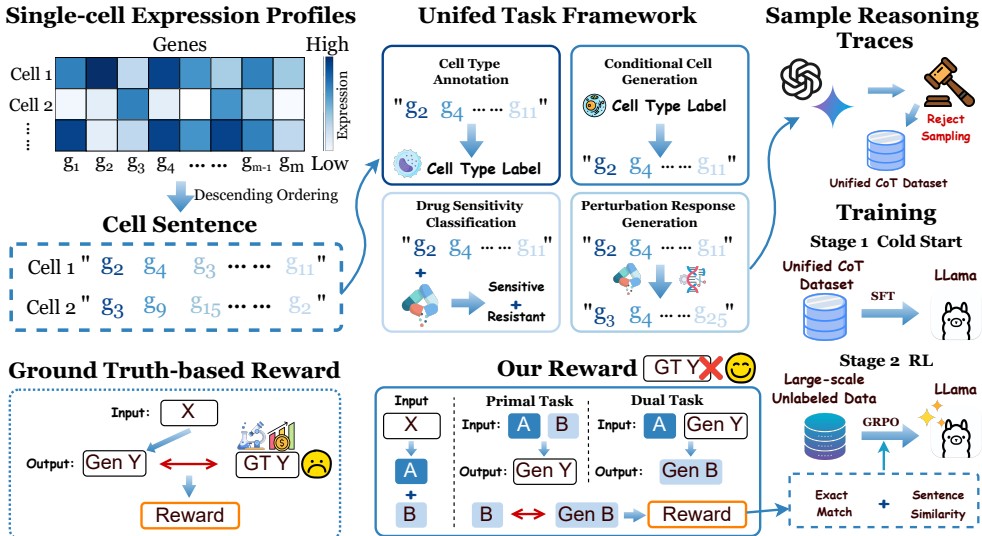

Figure 1: An overview of the `CellDuality` framework. Single-cell expression profiles are first converted into ranked "Cell Sentences," which are inputs to our task formulation covering four reasoning tasks. A high-quality CoT dataset is generated using a teacher model and Reject Sampling. This dataset is used for a Stage 1 SFT cold start of an LLaMA model. The model is then further aligned in Stage 2 via self-supervised RL (GRPO) on large-scale unlabeled data. The core innovation is our duality-based reward mechanism, which replaces the need for external Ground Truth by rewarding the consistency between a Primal Task and its complementary Dual Task.

achieve deep reasoning in a single task, while versatile, multi-task agents like InstructCell (Fang et al., 2025a) currently lack the same level of mechanistic insight. Therefore, developing a framework that takes a step toward deep reasoning across diverse tasks remains an open challenge. Such a framework must achieve generality over two core biological themes: cell identity and cell dynamics.

A promising direction is Reinforcement Learning from Verifiable Rewards (RLVR), a paradigm that has successfully enhanced LLMs' general reasoning ability, such as mathematics and code synthesis (Shao et al., 2024; Rafailov et al., 2023; Lee et al., 2023). However, its application in biology is severely limited because most biological reasoning tasks are inherently non-verifiable. For example, a specific gene sequence output of conditional cell generation has no single correct version for a given cell type, making simple verification infeasible. This data-dependency fundamentally constrains the training of more ambitious, unified models on the open-ended, cause-and-effect scenarios that would foster biological understanding.

To address this challenge, we introduce `CellDuality`, an agent for open-ended biological reasoning. It operates within a structured framework of four core tasks designed to span the fundamental biological themes of cell identity and cell dynamics (details in Sec. 3.1). Crucially, `CellDuality` is trained via a self-supervised paradigm inspired by DuPO (She et al., 2025), built on the principle of *Complementary Task Duality*. This framework leverages a bidirectional reasoning loop to generate its own supervisory signals: first, the model performs a forward reasoning task (e.g., predicting a cell's response to a drug); then, in a complementary inverse task, given generated results and known input conditions, it will reason backward to reconstruct the unknown input conditions. The reward is then determined by directly comparing the reconstructed input with the original. This consistency score becomes the intrinsic reward signal, compelling the model to produce forward predictions that are accurate and logically reversible, without needing ground-truth labels for the predictions themselves.

We implement this principle in a two-stage training paradigm. An initial Supervised Fine-Tuning (SFT) stage on a small, curated set of examples, containing both forward and inverse reasoning traces. This stage serves to cold-start the model, teaching it the language and format of biological reasoning. This is followed by a large-scale, self-supervised Reinforcement Learning (RL) stage, where the model is aligned using these intrinsic rewards on vast unlabeled data. This stage refines

the model's ability to produce outputs that are not only stylistically correct but also biologically and logically coherent.

Empirical evaluations demonstrate that `CellDuality`, despite being trained without any ground-truth verification during its RL phase, substantially outperforms its SFT-only counterpart. Critically, on the challenging OOD perturbation benchmark, our self-supervised approach closes 35-56% of the performance gap to a fully-supervised RLVR model that was trained with ground-truth rewards. This showcases the sample efficiency and generalization potential of our framework. Our main contributions are:

- We propose a structured framework that organizes complex biological inquiry into four core reasoning tasks spanning both cell identity and cell dynamics. This provides a promising step toward developing and evaluating agents with broader capabilities in single-cell biology.

- We introduce the principle of *Complementary Task Duality*, a new mechanism for generating annotation-free rewards. This framework incentivizes LLMs to learn the intrinsic mechanistic consistency of biological processes by rewarding the fidelity of a bidirectional reasoning loop, eliminating the need for ground-truth labels during the RL phase.

- We show empirically that the generalist model significantly outperforms standard SFT baselines and narrows the performance gap to a fully-supervised oracle model on the challenging out-of-distribution perturbation prediction benchmark.

## 2 RELATED WORK

**Foundation Models in Single-Cell Biology.** Foundation models are revolutionizing single-cell biology by learning representations from massive transcriptomic data (Cui et al., 2024; Theodoris et al., 2023). The field has rapidly progressed from models focused on predictive tasks, such as scGPT (Cui et al., 2024) for annotation and C2S-Scale (Rizvi et al., 2025) for multi-task generality, to those attempting explicit reasoning. However, these reasoning-aware models often operate in narrow paradigms; for instance, Cell-o1 (Fang et al., 2025b) frames reasoning as a logically-constrained puzzle, while agentic frameworks like ESCARGOT (Matsumoto et al., 2025) rely on external knowledge graphs. A key challenge remains in developing a single, generalist agent that can perform open-ended, mechanistic reasoning directly from cellular data. Our work addresses this gap, aiming for the generality of models like InstructCell (Fang et al., 2025a) but with a training objective that explicitly fosters deep, intrinsic reasoning.

**Reinforcement Learning from Verifiable Rewards .** Reinforcement learning is increasingly used to refine LLMs beyond standard SFT, with paradigms evolving to reduce reliance on external supervision (Ouyang et al., 2022; Shao et al., 2024). A particularly scalable paradigm is Reinforcement Learning from Verifiable Rewards (RLVR), which replaces subjective feedback (Bai et al., 2022) with objective, ground-truth-based rewards from deterministic verifiers (Shao et al., 2024). However, the prerequisite of a verifiable output severely limits RLVR's application in biology, where outcomes are inherently stochastic. Recent work has sought to address this by generating self-supervised rewards through task duality. For instance, the DuPO (She et al., 2025) framework introduced a generalized duality for non-invertible tasks, such as mathematical reasoning, by reconstructing input components to create a reward signal. Inspired by DuPO, our work adapts this duality principle to the unique challenges of biology (detailed in Sec. 3.2). We introduce Complementary Task Duality to generate intrinsic, self-verifiable rewards from the internal consistency of cellular processes, thus extending the RLVR paradigm to the non-verifiable biological domain.

## 3 METHODOLOGY

This section delineates the methodology for training our single-cell biological reasoning model. An overview of our entire framework is presented in Figure 1. We first define the concepts and the task formulation. We then introduce the generalized framework for self-supervision. Finally, we detail our training pipeline: an initial cold start stage, followed by a self-supervised Reinforcement Learning stage that uses our duality principle to enhance the model for deeper reasoning.

Table 1: Task Formulation for Single-Cell Reasoning.

| Theme | Classification Tasks | Generative Tasks |
|---|---|---|
| **Cell Identity** | **Cell Type Annotation**
(Input: Cell, Output: Label) | **Conditional Cell Generation**
(Input: Label, Output: Cell) |
| **Cell Dynamics** | **Drug Sensitivity Prediction**
(Input: Cell + Drug, Output: Label) | **Perturbation Response Generation**
(Input: Cell + Drug, Output: New Cell) |

## 3.1 PRELIMINARIES AND TASK FORMULATION

We denote the LLM policy as $\pi_\theta$, parameterized by $\theta$. Let $\mathcal{V}$ be the global vocabulary of all considered gene names. Our work addresses a structured set of four core single-cell reasoning tasks, organized into a 2x2 matrix spanning two fundamental biological themes: *Cell Identity* and *Cell Dynamics*. The core data structures for these tasks are defined as follows:

- **Cell Representation**: A cell $\mathbf{c} = \{g_1, g_2, \ldots, g_K\}$ is represented as an descending order sequence of its top $K$ expressed genes, where each gene $g_i \in \mathcal{V}$.

- **Perturbation**: A perturbation $\mathbf{p}$ is a structured tuple describing an intervention, e.g., $\mathbf{p} = \{\text{operation}, \text{target}\}$, where operation $\in \{\text{knockdown}, \text{overexpression}\}$ and target $\in \mathcal{V}$.

- **Cell Type and Sensitivity Labels**: A cell type $t$, is a categorical label from a predefined set $\mathcal{T}$. Similarly, a drug sensitivity label, $s$, is a categorical label from a set $\mathcal{S}$.

All inputs to the LLM are constructed as textual prompts $\mathbf{x}$ that combine these components. The model's output is a textual response $\mathbf{y}$, generated autoregressively according to the policy $\mathbf{y} \sim \pi_\theta(\cdot|\mathbf{x})$. A response may include a reasoning trace $\mathbf{z}$ and a final answer $\mathbf{a}$, i.e., $\mathbf{y} = \{\mathbf{z}, \mathbf{a}\}$.

## 3.2 THE PRINCIPLE OF COMPLEMENTARY TASK DUALITY

A primary obstacle to applying Reinforcement Learning (RL) to the four tasks defined above is the absence of a scalable reward source. In single-cell biology, obtaining ground-truth signals from experiments is prohibitively expensive and slow. Our work is motivated by a **central question**: Can we generate a reliable, intrinsic reward signal directly from the structure of these biological problems themselves, thus enabling RL without external supervision?

To achieve this, inspired by DuPO (She et al., 2025), we introduce a self-supervised reward generation framework. Adapting the duality principle from well-structured domains (e.g., mathematics) to biology is non-trivial: biological outputs are inherently stochastic and high-dimensional, requiring the design of domain-specific task formulations, such as conditional gene inpainting (Sec. 3.4), to produce stable reward signals. The core idea is to reframe a single biological question into a pair of mutually-verifying tasks, a primal task and a complementary dual task. This creates an internal logic loop that the model must satisfy, providing a natural source for an RL reward.

**Definition 3.1** (Complementary Task Duality). Let the input space $\mathcal{X}$ of a primal task $\mathcal{T}_p$ be decomposed into disjoint subspaces: $\mathcal{X}_k$ (known components) and $\mathcal{X}_u$ (unknown components), such that $\mathcal{X} = \mathcal{X}_k \cup \mathcal{X}_u$. The *primal task* $\mathcal{T}_p$ is a mapping from $\mathcal{T}_p : \mathcal{X} \to \mathcal{Y}$. Its *complementary dual task* $\mathcal{T}_{cd}$ is a mapping that leverages the primal output $\mathbf{y}$ and the known component $\mathbf{x}_k$ to reconstruct the unknown component $\hat{\mathbf{x}}_u$:

$$\mathcal{T}_{cd} : (\mathbf{y}, \mathbf{x}_k) \mapsto \hat{\mathbf{x}}_u.$$

Pair $(\mathcal{T}_p, \mathcal{T}_{cd})$ forms a *generalized dual pair* if it satisfies the *complementary consistency principle*:

$$\forall \mathbf{x} \in \mathcal{X}, \mathbf{y} = \mathcal{T}_p(\mathbf{x}) : \quad d\big(\mathbf{x}_u, \mathcal{T}_{cd}(\mathbf{y}, \mathbf{x}_k)\big) \leq \epsilon,$$

where $d(\cdot, \cdot) : \mathcal{X}_u \times \mathcal{X}_u$ is a domain-specific distance metric, and $\epsilon \geq 0$ is a tolerance threshold.

The power of this framework lies in its ability to transform an unsupervised problem into a self-verifying one. The consistency principle defined above provides the mechanism to generate rewards: the fidelity of the dual task's reconstruction, $d(\mathbf{x}_u, \hat{\mathbf{x}}_u)$, serves as a direct, intrinsic measure of the logical and biological coherence of the primal task's output $\mathbf{y}$. This approach elegantly sidesteps the challenges of classical dual learning (irreversibility and asymmetry) by leveraging the known component $\mathbf{x}_k$ as a contextual anchor, ensuring the dual task is well-posed.

## 3.3 Training Stage 1: Supervised Fine-Tuning for Capability Cold-Start

Before RL Training, we first initialize the base LLM with foundational biological knowledge and reasoning patterns through Supervised Fine-Tuning (SFT). This essential cold-start phase ensures the model can effectively engage with the complex, self-supervised tasks in the subsequent alignment stage. The process involves two key steps: generating a high-quality Chain-of-Thought dataset, and then using it to train the model.

### 3.3.1 Chain-of-Thought Reasoning Dataset Generation

We construct a comprehensive SFT dataset, $\mathcal{D}_{\text{SFT}}$, by leveraging powerful teacher models (e.g., GPT-4o, Gemini 2.5 Pro) to generate Chain-of-Thought (CoT) reasoning traces. A critical aspect of our approach is that $\mathcal{D}_{\text{SFT}}$ must equip our model with capabilities for both the primal (forward) reasoning and the complementary dual (inverse) reasoning required in our RL stage. Therefore, we generate and curate distinct data subsets for each direction.

**Primal Task SFT Data.** For each of our four core tasks, we generate primal task data. Given an input prompt $\mathbf{x}_i$, we prompt a teacher model $\pi_{\text{teacher}}$ to generate $N$ candidate responses $\{\mathbf{y}_{i,k} = (\mathbf{z}_{i,k}, \mathbf{a}_{i,k})\}_{k=1}^N$. We then apply task-specific filtering to select high-quality instances for our primal SFT set, $\mathcal{D}_{\text{SFT}}^{\text{primal}}$.

- **For Classification Tasks (Annotation & Sensitivity)**: We use a strict *Rejection Sampling* protocol. A candidate $\mathbf{y}_{i,k}$ is accepted if its final answer $\mathbf{a}_{i,k}$ exactly matches the ground-truth label $\mathbf{a}_i^*$. We define an indicator for correctness as $\epsilon_{i,k} = \mathbb{I}(\mathbf{a}_{i,k} = \mathbf{a}_i^*)$. The accepted set for prompt $\mathbf{x}_i$ is $\{\mathbf{y}_{i,k}|\epsilon_{i,k} = 1\}$. This ensures all training examples are factually correct.
- **For Generative Tasks (Cell & Response Generation)**: As no single, unique ground-truth sequence exists for these tasks, a simple exact match is infeasible. Instead, we adopt a *Rank-Aware Filtering* protocol. For each prompt $\mathbf{x}_i$ with a corresponding ground-truth cell sequence $\mathbf{a}_i^*$, the teacher model generates a candidate response $(\mathbf{z}_i, \mathbf{a}_i)$. The candidate is accepted into $\mathcal{D}_{\text{SFT}}$ only if the generated cell sequence $\mathbf{a}_i$ demonstrates high fidelity to the ground truth in terms of both gene overlap and expression ranking. We quantify this using our proposed Rank-Weighted Jaccard Similarity metric (Eq. 3). A candidate is accepted only if its similarity score exceeds a predefined threshold.

**Dual Task SFT Data.** To explicitly teach the model the inverse reasoning required for our duality framework, we construct a corresponding dual task SFT set, $\mathcal{D}_{\text{SFT}}^{\text{dual}}$. For each instance in our curated primal set, we formulate its complementary dual problem.

- For a primal instance $(\mathbf{x} = (\mathbf{x}_k, \mathbf{x}_u), \mathbf{y}^*)$, we construct a dual prompt $\mathbf{x}_{\text{dual}} = (\mathbf{y}^*, \mathbf{x}_k)$. The ground-truth answer for this dual task is the original unknown component, $\mathbf{y}_{\text{dual}}^* = \mathbf{x}_u$.
- For example, for a perturbation response instance where $\mathbf{x}_k = \mathbf{c}_{\text{pre}}$, $\mathbf{x}_u = \mathbf{p}$, and $\mathbf{y}^* = \mathbf{c}_{\text{post}}^*$, the dual SFT sample would be: prompt $(\mathbf{c}_{\text{post}}^*, \mathbf{c}_{\text{pre}})$ paired with the ground-truth answer $\mathbf{p}$.

The teacher model is then prompted to generate CoT reasoning for these dual problems. The final SFT dataset is the union $\mathcal{D}_{\text{SFT}} = \mathcal{D}_{\text{SFT}}^{\text{primal}} \cup \mathcal{D}_{\text{SFT}}^{\text{dual}}$. This hybrid strategy ensures the model is proficient in both forward and inverse reasoning before entering the RL stage.

### 3.3.2 Supervised Fine-tuning Objective

The model is then trained on $\mathcal{D}_{\text{SFT}}$ by minimizing the standard negative log-likelihood loss $\mathcal{L}_{\text{SFT}}(\theta)$ over the complete reasoning trajectories:

$$\mathcal{L}_{\text{SFT}}(\theta) = -\mathbb{E}_{(\mathbf{x}_i, \mathbf{y}_i^*) \sim \mathcal{D}_{\text{SFT}}} \left[ \sum_{j=1}^{|\mathbf{y}_i^*|} \log \pi_\theta(y_{i,j}^* | \mathbf{x}_i, \mathbf{y}_{i,<j}^*) \right]. \tag{1}$$

The resulting model, $\pi_{\text{SFT}}$, possesses the baseline capabilities required for the subsequent self-supervised alignment.

## 3.4 Stage 2: Self-Supervised Duality-Guided Reinforcement Learning

This stage constitutes the core of our self-supervised methodology. We refine the capabilities of the SFT-initialized model, $\pi_{\text{SFT}}$, by aligning it with the principle of complementary consistency. This is achieved through a Reinforcement Learning (RL) framework that operates on a large, unlabeled dataset $\mathcal{D}_{\text{RL}}$ and is guided by intrinsic rewards, eliminating the need for any ground-truth data.

### 3.4.1 Self-Supervised Reward Generation

The cornerstone of our alignment stage is the generation of intrinsic rewards derived from the complementary duality principle. For any prompt $\mathbf{x} = (\mathbf{x}_k, \mathbf{x}_u)$ and a model-generated primal output $\mathbf{y}$, we compute a reward by executing the complementary dual task and measuring its reconstruction fidelity. We employ two types of rewards, categorical and sequence-based, tailored to the nature of our core tasks.

**Categorical Rewards from Inverse Task Consistency.** For generative tasks such as Perturbation Response Generation and Conditional Cell Generation, the duality provides a clean, categorical reward signal. In both cases, the primal task generates a high-dimensional cell sequence ($\mathbf{c}_{\text{post}}$ or $\mathbf{c}$), and the complementary dual task attempts to reconstruct a categorical input label (the sensitivity $\mathbf{s}$ or the cell type $t$). The reward is a binary signal based on the exact reconstruction of this label:

$$r(\mathbf{y}|\mathbf{x}) = \mathbb{I}(\hat{\mathbf{x}}_u = \mathbf{x}_u), \tag{2}$$

where $\mathbf{x}_u$ is the original categorical input (e.g., $t$) and $\hat{\mathbf{x}}_u$ is its reconstruction (e.g., $\hat{t}$). This reward directly measures the logical consistency of the generated output: a biologically plausible cell sequence should unambiguously encode the conditions that generated it.

**Continuous Rewards from Conditional Inpainting.** For classification tasks such as Cell Type Annotation and Drug Sensitivity Prediction, where the primal output is a low-information label, we design a reward based on a conditional gene inpainting objective. Here, the input cell sequence is artificially decomposed into an observed part $\mathbf{c}_{\text{obs}}$ and a hidden part $\mathbf{c}_{\text{hid}}$, which serves as the unknown component $\mathbf{x}_u$. The dual task is to reconstruct $\hat{\mathbf{c}}_{\text{hid}}$ conditioned on both the observed genes $\mathbf{c}_{\text{obs}}$ and the model's predicted primal label ($\hat{t}$ or $\hat{s}$). The reward is a continuous score reflecting the quality of this reconstruction: $r(\hat{t}|\mathbf{c}) = \text{RWJS}(\mathbf{c}_{\text{hid}}, \hat{\mathbf{c}}_{\text{hid}})$. The Rank-Weighted Jaccard Similarity (RWJS) extends the standard Jaccard index by weighting each gene by its reciprocal rank, $w(g, \mathbf{c}) = 1/\text{rank}(g, \mathbf{c})$, so that higher-expressed genes contribute more. For a ground-truth sequence $\mathbf{c}^*$ and a generated sequence $\mathbf{c}_{\text{gen}}$, it is defined as:

$$\text{RWJS}(\mathbf{c}^*, \mathbf{c}_{\text{gen}}) = \frac{\sum_{g \in S^* \cap S_{\text{gen}}} \frac{w(g, \mathbf{c}^*) + w(g, \mathbf{c}_{\text{gen}})}{2}}{\sum_{g \in S^*} w(g, \mathbf{c}^*) + \sum_{g \in S_{\text{gen}} \setminus S^*} w(g, \mathbf{c}_{\text{gen}})}, \tag{3}$$

where $S^* = \text{Set}(\mathbf{c}^*)$ and $S_{\text{gen}} = \text{Set}(\mathbf{c}_{\text{gen}})$. RWJS ranges from 0 (no overlap) to 1 (identical), providing a biologically meaningful measure that prioritizes high-expression genes. This reward incentivizes the model to base its classification on a deep understanding of the cell's underlying gene signature, as a correct label should provide the necessary context for accurate gene inpainting.

### 3.4.2 Policy Optimization with GRPO

We optimize the policy $\pi_\theta$ to maximize the expected self-supervised reward $\mathcal{J}(\theta) = \mathbb{E}_{\mathbf{x} \sim \mathcal{D}_{\text{RL}}}[r(\mathbf{y}|\mathbf{x})]$. We employ Group Relative Policy Optimization (GRPO), a memory-efficient and stable critic-free RL algorithm. The optimization follows an iterative, online process: for each prompt, the current policy $\pi_\theta$ generates a group of $G$ candidate responses, each of which is then assigned a self-supervised reward based on its dual-task performance. This group of responses and rewards is then used to update the policy as follows.

**Advantage Estimation.** For each prompt, after generating a group of $G$ responses and their corresponding rewards $\{r_k\}_{k=1}^{G}$, we compute the advantage for each candidate. This is achieved by normalizing the rewards relative to the group's performance, which serves as an empirical baseline, thus obviating the need for a separate value function:

$$A_k = \frac{r_k - \text{mean}(\{r_j\}_{j=1}^{G})}{\text{std}(\{r_j\}_{j=1}^{G}) + \epsilon}. \tag{4}$$

Table 2: Performance comparison on the Cell Type Annotation task. Baselines are trained on each dataset individually. We report Accuracy (Acc.) and Macro F1-score (F1) for all five benchmarks, differentiating between ID and OOD evaluation for our model.

| Model | In-Distribution (ID) Evaluation | | | | | | Out-of-Distribution (OOD) Evaluation | | | |
| | He-2020-Liver | | Segerstolpe-2016 | | Xin-2016 | | Ma-2020 | | Bastidas-Ponce-2019 | |
| | Acc. (%) | F1 (%) | Acc. (%) | F1 (%) | Acc. (%) | F1 (%) | Acc. (%) | F1 (%) | Acc. (%) | F1 (%) |
|---|---|---|---|---|---|---|---|---|---|---|
| scBERT | 95.28 | 94.08 | 99.52 | 99.64 | 99.25 | 98.79 | 82.92 | 81.73 | 86.67 | 79.60 |
| scGPT | 94.88 | 91.75 | 98.09 | 97.82 | 99.10 | 98.40 | 82.84 | 79.40 | **91.43** | 87.01 |
| Geneformer | 96.06 | 92.57 | 99.52 | 99.49 | **99.70** | **99.39** | **85.79** | **84.89** | 88.50 | 83.81 |
| Cell2Sentence | 94.88 | 94.42 | 99.52 | 99.64 | 99.35 | 98.77 | 82.40 | 81.05 | 80.59 | 76.82 |
| InstructCell-instruct | 96.06 | 95.24 | **100.00** | **100.00** | 99.30 | 98.89 | 85.59 | 84.56 | 91.10 | **88.69** |
| CellDuality (SFT-only) | 94.83$_{\pm0.21}$ | 94.67$_{\pm0.18}$ | 98.76$_{\pm0.08}$ | 98.73$_{\pm0.09}$ | 99.45$_{\pm0.12}$ | 99.01$_{\pm0.15}$ | 80.22$_{\pm0.34}$ | 74.95$_{\pm0.41}$ | 72.87$_{\pm0.28}$ | 57.24$_{\pm0.33}$ |
| CellDuality | **96.34**$_{\pm0.19}$ | **95.41**$_{\pm0.16}$ | 99.81$_{\pm0.07}$ | 99.78$_{\pm0.08}$ | 99.52$_{\pm0.11}$ | 99.08$_{\pm0.13}$ | 82.03$_{\pm0.32}$ | 81.78$_{\pm0.39}$ | 88.45$_{\pm0.26}$ | 78.12$_{\pm0.31}$ |

**Objective Function.** The policy is updated by maximizing the GRPO objective, which includes a clipped surrogate objective to stabilize training and a KL penalty to prevent large deviations from a reference policy $\pi_{\text{ref}}$ (typically the initial SFT model $\pi_{\text{SFT}}$):

$$\mathcal{J}_{\text{GRPO}}(\theta) = \mathbb{E}\left[\min\left(\rho_t(\theta)A_t, \text{clip}(\rho_t(\theta), 1 - \epsilon_c, 1 + \epsilon_c)A_t\right) - \beta D_{\text{KL}}(\pi_\theta \| \pi_{\text{ref}})\right], \quad (5)$$

where $\rho_t(\theta) = \pi_\theta(y_t|x)/\pi_{\theta_{\text{old}}}(y_t|x)$ is the probability ratio, $A_t$ is the advantage at token $t$ (in our case, $A_k$ is applied to all tokens of response $k$), $\epsilon_c$ is the clipping ratio, and $\beta$ is the KL coefficient. This iterative, multi-task training process progressively refines the model's ability to generate biologically coherent and logically consistent responses.

## 4 EXPERIMENT

### 4.1 EXPERIMENTAL SETUP

**Tasks and Datasets.** Our evaluation is centered around the four core single-cell reasoning tasks introduced in our framework. To ensure fair and direct comparison with state-of-the-art models, we adopt the exact datasets and train/test splits used in seminal works, including C2S-Scale (Rizvi et al., 2025) and InstructCell (Fang et al., 2025a). Our training strategy involves fine-tuning a single base model on a designated primary training set for each task theme, and then evaluating its performance on both in-distribution (ID) and out-of-distribution (OOD) test sets.

- **Cell Identity Tasks (Annotation & Generation):**
  - **Training Dataset:** To build a robust multi-task model for cell identity, we construct a mixed training dataset by combining the training splits of four diverse public benchmarks: He-2020-Liver (He et al., 2020), Segerstolpe-2016 (Segerstolpe et al., 2016), Xin-2016 (Xin et al., 2016), and Human Immune Tissue Dataset (Domínguez Conde et al., 2022). This mixed dataset serves as the sole source of supervision for our model on all identity tasks.
  - **ID Test Set:** For the cell type annotation task, we use the held-out test splits of the three datasets included in our training mix (He-2020-Liver, Segerstolpe-2016, Xin-2016). For cell generation, we use the held-out test splits of Human Immune Datasets.
  - **OOD Test Set:** We use two datasets entirely unseen during training: Ma-2020 (Ma et al., 2020) and Bastidas-Ponce-2019 (Bastidas-Ponce et al., 2019).
- **Cell Dynamics Tasks (Sensitivity Prediction & Response Generation):**
  - **Training Dataset:** We construct a comprehensive mixed training dataset by combining three distinct perturbation benchmarks: the L1000 dataset (Subramanian et al., 2017), which covers two human drug response datasets, GSE149383 (Lung) (Aissa et al., 2021) and GSE117872 (Oral Cavity) (Sharma et al., 2018). This diverse dataset, containing examples for both response generation and sensitivity classification, serves as the sole source of supervision for our model on all dynamics-related tasks.
  - **ID Test Sets:** The held-out test splits of the two datasets explicitly included for the classification task: GSE149383 and GSE117872.
  - **OOD Test Sets:** We use two benchmarks entirely unseen during training. For cross-species classification, we use the complete GSE110894 (Mouse Bone Marrow) dataset (Bell et al., 2019). For generative causal reasoning, we use the OOD splits of the sci-Plex3 Human Perturbation dataset (Srivatsan et al., 2020).

Table 3: Performance comparison on the Drug Sensitivity Classification task. Baselines are trained on each dataset individually. We report Accuracy (Acc.) and Macro F1-score (F1).

| Model | In-Distribution (ID) Evaluation | | | | Out-of-Distribution (OOD) Evaluation | |
|---|---|---|---|---|---|---|
| | GSE149383 (Human Lung) | | GSE117872 (Human Oral) | | GSE110894 (Mouse Bone) | |
| | Acc. (%) | F1 (%) | Acc. (%) | F1 (%) | Acc. (%) | F1 (%) |
| scBERT | **99.56** | **99.56** | 95.42 | 96.01 | 95.80 | 95.79 |
| scGPT | 97.79 | 97.79 | 82.44 | 84.76 | 95.80 | 95.79 |
| Geneformer | 98.23 | 98.23 | 94.66 | 95.27 | 93.01 | 92.91 |
| Cell2Sentence | 93.36 | 93.36 | 90.84 | 90.72 | 95.10 | 95.08 |
| InstructCell-instruct | 97.35 | 97.34 | **100.00** | **100.00** | **97.20** | **97.19** |
| `CellDuality` (SFT-only) | $98.91_{\pm0.15}$ | $98.89_{\pm0.16}$ | $96.78_{\pm0.22}$ | $97.12_{\pm0.19}$ | $\underline{96.45}_{\pm0.18}$ | $\underline{96.42}_{\pm0.20}$ |
| `CellDuality` | $\underline{99.12}_{\pm0.13}$ | $\underline{99.10}_{\pm0.14}$ | $\underline{97.23}_{\pm0.20}$ | $\underline{97.58}_{\pm0.17}$ | $96.12_{\pm0.21}$ | $96.08_{\pm0.23}$ |

Table 4: Performance on Perturbation Response Generation (sci-Plex3 benchmark). Our model was trained on a separate perturbation dataset, while baselines were trained on the in-distribution splits of sci-Plex3. Lower scores are better for distribution-based metrics.

| Model | Supervision Type | scFID ($\downarrow$) | MMD ($\downarrow$) | Wasserstein ($\downarrow$) |
|---|---|---|---|---|
| scGen | Supervised | 0.95 | 1.05 | 0.98 |
| CellOT | Supervised | 0.88 | 1.03 | 0.95 |
| scGPT | Supervised | 0.29 | 0.42 | 0.54 |
| C2S-Scale 1B (SFT) | Supervised | **0.02** | **0.01** | 0.21 |
| C2S-Scale (GRPO w/ GT) | **Ground-Truth RL** | **0.02** | **0.01** | **0.21** |
| `CellDuality` (SFT-only) | Supervised | $0.045_{\pm0.003}$ | $0.028_{\pm0.002}$ | $0.267_{\pm0.012}$ |
| `CellDuality` | **Self-Supervised** | $\underline{0.038}_{\pm0.002}$ | $\underline{0.019}_{\pm0.001}$ | $\underline{0.245}_{\pm0.011}$ |

**Evaluation Metrics.**

- For Classification Tasks (Cell Type Annotation, Drug Sensitivity): We report *Accuracy* as the primary metric. We also include the *Macro F1-score* to account for class imbalance.

- For Generative Tasks (Conditional Generation, Perturbation Response): For Perturbation Response Generation, we follow C2S-Scale (Rizvi et al., 2025) and report distribution-based metrics (scFID and MMD) calculated in a pre-trained embedding space (scGPT (Cui et al., 2024)) to assess the quality and realism of generated cell populations. For Conditional Cell Generation, we follow Cell2Sentence (LeVine et al., 2024) and report Gromov-Wasserstein (GW) Distance and k-NN Accuracy. The k-NN classifier is evaluated with multiple neighbor values ($k \in \{3, 5, 10, 25\}$).

**Baseline Models.** We benchmark `CellDuality` against a comprehensive set of state-of-the-art models, with all performance metrics cited directly from the original publications for fair comparison. For classification tasks (Annotation and Sensitivity), we compare against domain-specific foundation models such as **scGPT** (Cui et al., 2024) and **Geneformer** (Theodoris et al., 2023), as well as LLM-based agents like **InstructCell** (Fang et al., 2025a). For generative tasks (Cell and Response Generation), baselines include specialized generative models like **scGen** (Lotfollahi et al., 2019) and **scDiffusion** (Luo et al., 2024), and the powerful LLM-based framework **C2S-Scale** (Rizvi et al., 2025).

**Implementation Details.** Our `CellDuality` model is based on the Llama-3.2-3B architecture. The SFT stage is conducted for 3 epochs with a learning rate of $1e-5$. The subsequent self-supervised RL alignment is performed using GRPO with a group size of $G = 8$, a train batch size of 512, a mini-batch size of 32, and is run for 200 optimization steps. All experiments are conducted on 8x A6000 GPUs. All our scores are shown as mean $\pm$ std over 5 runs.

## 4.2 Main Results

Across all four reasoning tasks, our self-supervised framework, `Cell-Duality`, demonstrates highly competitive performance against a wide range of state-of-the-art baselines. As detailed in Tables 2 through 5, our multi-task model, trained on mixed datasets, consistently matches or surpasses specialist models that were trained on individual benchmarks. This is particularly evident in the classification tasks (Cell Type Annotation and Drug Sensitivity), where `CellDuality` shows robust generalization to out-of-distribution and even cross-species datasets.

Table 5: Performance on Conditional Cell Generation on the Human Immune dataset (In-Distribution). Baseline results are cited from Cell2Sentence (LeVine et al., 2024). k-NN Accuracy is reported for multiple values of k.

| Model | k-NN Accuracy (%) ↑ | | | | GW Distance (↓) |
|---|---|---|---|---|---|
| | k=3 | k=5 | k=10 | k=25 | |
| scVI | $24.36_{\pm 0.0062}$ | $24.00_{\pm 0.0064}$ | $24.25_{\pm 0.0034}$ | $23.48_{\pm 0.0032}$ | $302.13_{\pm 0.9338}$ |
| scGen | $23.76_{\pm 0.0112}$ | $23.30_{\pm 0.0093}$ | $23.77_{\pm 0.0053}$ | $23.35_{\pm 0.0041}$ | $315.95_{\pm 1.2431}$ |
| scDiffusion | $23.35_{\pm 0.0125}$ | $22.88_{\pm 0.0111}$ | $23.68_{\pm 0.0067}$ | $23.06_{\pm 0.0049}$ | $72.02_{\pm 0.3937}$ |
| scGPT | $18.38_{\pm 0.0086}$ | $17.88_{\pm 0.0169}$ | $18.11_{\pm 0.0149}$ | $18.82_{\pm 0.0071}$ | $2989.81_{\pm 4.9229}$ |
| Cell2Sentence-160M | $\underline{25.88}_{\pm 0.0061}$ | $\underline{25.65}_{\pm 0.0060}$ | $\mathbf{27.46}_{\pm 0.0073}$ | $\mathbf{27.15}_{\pm 0.0070}$ | $\mathbf{54.30}_{\pm 0.3410}$ |
| CellDuality (SFT-only) | $24.92_{\pm 0.0058}$ | $24.71_{\pm 0.0055}$ | $25.83_{\pm 0.0062}$ | $25.49_{\pm 0.0059}$ | $63.87_{\pm 0.0421}$ |
| CellDuality | $\mathbf{26.34}_{\pm 0.0056}$ | $\mathbf{25.92}_{\pm 0.0053}$ | $\underline{26.21}_{\pm 0.0060}$ | $\underline{25.98}_{\pm 0.0057}$ | $\underline{61.45}_{\pm 0.0408}$ |

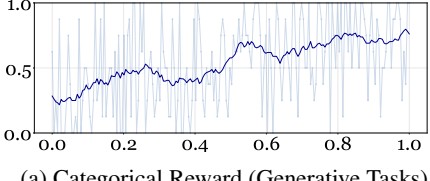

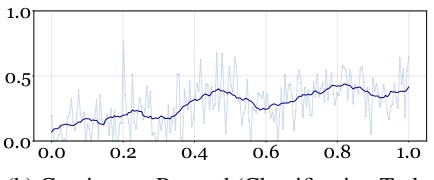

(a) Categorical Reward (Generative Tasks)      (b) Continuous Reward (Classification Tasks)

Figure 2: **Training dynamics of self-supervised rewards during the RL alignment stage.** The plots show the moving average of (a) the categorical accuracy-based reward for generative tasks and (b) the continuous RWJS-based reward for classification tasks.

The most significant impact of our self-supervised approach is observed in the generative tasks requiring deep mechanistic reasoning. For Perturbation Response Generation, the duality-guided RL stage provides a substantial performance boost over the already strong SFT baseline. Critically, our self-supervised model successfully narrows the performance gap to a fully-supervised oracle that requires ground-truth labels for alignment, proving the efficacy of our annotation-free strategy. While our model also demonstrates strong performance on Conditional Cell Generation by outperforming most classical and deep learning-based generative models, its primary strength lies in its ability to learn the intrinsic, mechanistic consistency of biological processes, showcasing a new path toward scalable and robust scientific reasoning agents. The central value of our self-supervised approach lies in eliminating the dependence on expensive, often unavailable ground-truth labels during the RL alignment phase. Our framework achieves competitive performance without any such supervision, providing a practical and scalable path to strong reasoning capabilities in data-scarce biological domains.

## 4.3 ABLATION STUDY

**Self-Supervised vs. Ground-Truth Supervised RL** To rigorously quantify the efficacy of our self-supervised alignment strategy, we conduct a head-to-head comparison against a standard supervised RL approach. We evaluate three key models on their respective in-distribution test sets: (1) the SFT-only baseline, (2) a supervised RL oracle trained with ground-truth rewards, and (3) our self-supervised CellDuality model. As shown in Table 6, our self-supervised RL approach consistently and significantly boosts performance over the SFT-only baseline across all tasks. Critically, our annotation-free method substantially narrows the performance gap to the fully-supervised oracle, and even surpasses the oracle's Macro F1-score on the He-2020-Liver annotation task, suggesting it learns more robust decision boundaries.

## 5 CONCLUSION

We introduced CellDuality, an agent that learns complex biological reasoning through a novel self-supervised framework. Our core contribution, the principle of complementary task duality, enables reinforcement learning alignment on non-verifiable single-cell tasks by generating intrinsic rewards from a bidirectional reasoning loop. Trained via our sample-efficient, two-stage paradigm,

Table 6: Core ablation study comparing Self-Supervised RL against a Ground-Truth Supervised oracle. All models are initialized from the same SFT checkpoint and evaluated on their respective **in-distribution (ID)** test sets.

| Method Configuration | He-2020-Liver | | GSE149383 (Lung) | | sci-Plex3 | |
|---|---|---|---|---|---|---|
| | Acc. ($\uparrow$) | F1 ($\uparrow$) | Acc. ($\uparrow$) | F1 ($\uparrow$) | scFID ($\downarrow$) | MMD ($\downarrow$) |
| Llama-3.2-3B-Instruct | $22.45_{\pm1.23}$ | $52.82_{\pm1.45}$ | $29.67_{\pm0.89}$ | $61.34_{\pm1.12}$ | - | - |
| SFT-only | $95.83_{\pm0.21}$ | $94.67_{\pm0.18}$ | $98.91_{\pm0.15}$ | $98.89_{\pm0.16}$ | $0.045_{\pm0.003}$ | $0.028_{\pm0.002}$ |
| RL with Ground-Truth | $\mathbf{97.21}_{\pm0.16}$ | $\underline{94.85}_{\pm0.14}$ | $\mathbf{99.34}_{\pm0.12}$ | $\mathbf{99.31}_{\pm0.13}$ | $\mathbf{0.025}_{\pm0.001}$ | $\mathbf{0.012}_{\pm0.001}$ |
| **Ours (Self-Supervised RL)** | $\underline{96.34}_{\pm0.19}$ | $\mathbf{95.41}_{\pm0.16}$ | $\underline{99.12}_{\pm0.13}$ | $\underline{99.10}_{\pm0.14}$ | $\underline{0.038}_{\pm0.002}$ | $\underline{0.019}_{\pm0.001}$ |

`CellDuality` achieves state-of-the-art performance across four distinct reasoning tasks, providing coherent biological explanations. Critically, our self-supervised approach demonstrates its efficacy by narrowing the performance gap to a supervised RLVR baseline. This work presents a promising step toward scalable foundation models in biology, offering a new paradigm that learns to reason from the intrinsic logical structure of scientific problems, rather than from external labels.

## LIMITATIONS

Our framework is validated on four representative transcriptomic tasks and has not yet been extended to other omics modalities such as ATAC-seq or proteomics. The cell sentence representation discards absolute expression magnitudes, trading encoding precision for compatibility with off-the-shelf LLMs. While our duality-based reward enforces logical consistency, it does not guarantee biological correctness. Mutually consistent yet biologically implausible predictions remain possible, and direct interpretability evaluation at the pathway level is left for future work. The RL gains are most pronounced for generative tasks and comparatively modest for classification tasks where SFT alone already achieves strong performance. Finally, our experiments use a 3B-parameter model, which may limit reasoning depth. Exploring the scaling behavior of our approach is an important future direction.

## ACKNOWLEDGEMENTS

This research was partially funded by the National Institutes of Health (NIH) under award 1R01EB03710101. The views and conclusions contained in this document are those of the authors and should not be interpreted as representing the official policies, either expressed or implied, of the NIH. This research was also partially supported by the Amazon Research Award.

## THE USE OF LARGE LANGUAGE MODELS

We utilized Google's Gemini Pro 2.5 as a writing assistant in the preparation of this manuscript. Its function was strictly limited to language refinement tasks, such as enhancing clarity, correcting grammar, and rephrasing sentences to improve readability within an academic context. All scientific content, including the core ideas, experimental design, and interpretation of results, was generated exclusively by the human authors.

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
