# APPENDIX CONTENTS

## A  RANK-WEIGHTED JACCARD SIMILARITY

The Rank-Weighted Jaccard Similarity (RWJS) extends the traditional Jaccard Similarity by assigning a weight to each element based on its position (rank) in the sequences. This ensures that genes with higher expression (appearing earlier in the sequence) contribute more significantly to the similarity score, aligning the metric more closely with biological importance.

Given a ground-truth cell sequence $\mathbf{c}^* = (g_1^*, g_2^*, \ldots, g_K^*)$ and a generated cell sequence $\mathbf{c}_{\text{gen}} = (g_1, g_2, \ldots, g_K)$, the RWJS is calculated as the ratio of the weighted intersection to the weighted union.

**1. Rank-based Weighting.** First, we define a weight function $w(g, \mathbf{c})$ for a gene $g$ in a sequence $\mathbf{c}$. We use the reciprocal rank as the weight, which heavily favors higher-ranked items:

$$w(g, \mathbf{c}) = \frac{1}{\text{rank}(g, \mathbf{c})}, \tag{5}$$

where $\text{rank}(g, \mathbf{c})$ is the position of gene $g$ in sequence $\mathbf{c}$ (indexed from 1). If a gene is not in the sequence, its rank can be considered infinite and its weight zero.

**2. Weighted Intersection.** The weighted intersection, $I_w$, considers only the genes present in both sequences. For each common gene, we take the average of its weights from both sequences to provide a balanced contribution:

$$I_w(\mathbf{c}^*, \mathbf{c}_{\text{gen}}) = \sum_{g \in \text{Set}(\mathbf{c}^*) \cap \text{Set}(\mathbf{c}_{\text{gen}})} \frac{w(g, \mathbf{c}^*) + w(g, \mathbf{c}_{\text{gen}})}{2}. \tag{6}$$

**3. Weighted Union.** The weighted union, $U_w$, is the sum of weights of all unique genes across both sequences. This is calculated by summing the weights of all genes in the ground-truth sequence and adding the weights of any genes that are unique to the generated sequence:

$$U_w(\mathbf{c}^*, \mathbf{c}_{\text{gen}}) = \sum_{g \in \text{Set}(\mathbf{c}^*)} w(g, \mathbf{c}^*) + \sum_{g \in \text{Set}(\mathbf{c}_{\text{gen}}) \setminus \text{Set}(\mathbf{c}^*)} w(g, \mathbf{c}_{\text{gen}}). \tag{7}$$

**4. Final RWJS Score.** The final RWJS score is the ratio of the weighted intersection to the weighted union, bounded between 0 and 1:

$$\text{RWJS}(\mathbf{c}^*, \mathbf{c}_{\text{gen}}) = \frac{I_w(\mathbf{c}^*, \mathbf{c}_{\text{gen}})}{U_w(\mathbf{c}^*, \mathbf{c}_{\text{gen}})}. \tag{8}$$

A score of 1 indicates identical sequences, while a score of 0 indicates no common genes. This metric provides a nuanced assessment of generative quality, rewarding models that correctly identify high-expression genes and place them in their proper high-ranking positions.

## B  THE USE OF LARGE LANGUAGE MODELS (LLMS)

We utilized Google's Gemini Pro 2.5 as writing assistants in the preparation of this manuscript. Their function was strictly limited to language refinement tasks, such as enhancing clarity, correcting grammar, and rephrasing sentences to improve readability within an academic context. All scientific content, including the core ideas, experimental design, and interpretation of results, was generated exclusively by the human authors. This use of AI for language polishing is in accordance with standard academic and ICLR guidelines.