# OpenReview forum: "CellDuality: Unlocking Biological Reasoning in LLMs with Self-Supervised RLVR"
_ICLR.cc/2026/Conference — ICLR 2026 Poster_

### Official Review · Reviewer_nVH5 · 2025-10-29

**Soundness:** 3
**Presentation:** 3
**Contribution:** 2
**Rating:** 2
**Confidence:** 2

**Summary:**

This paper proposes CellDuality, a self-supervised RL-framework that enables large language models (LLMs) to perform open-ended biological reasoning in single-cell analysis. The key idea is to make sure each biological reasoning task such as drug response prediction is paired with an inverse task such as  recovering perturbation from response, forming a bidirectional reasoning loop. The consistency between the forward and inverse predictions is used as an intrinsic reward signal without labeling or annotation, enabling RL alignment without ground-truth verification labels. Experiments across four representative single-cell reasoning tasks demonstrate that CellDuality achieves competitive or slightly better performance compared to SFT baselines and narrows the gap to fully label-supervised RLVR models.

**Strengths:**

- The proposed self-supervised RL training paradigm is training-efficient and agnostic to the choice of policy optimization algorithm, showing that consistent rewards can provide alignment signals without external supervision.
- The Complementary Task Duality concept is clearly formulated and grounded in prior work (eg., DuPO), providing a natural extension to domains where canonical verifiable rewards are not available or expensive to acquire.

**Weaknesses:**

- The self-consistency reward effectively measures semantic or reconstruction fidelity rather than true biological correctness. While this provides a stable and well-defined signal, it may not inject new knowledge or enable deeper reasoning as canonical verifiers do in math, code, or symbolic domains. In this sense, the biological reasoning learned may remain shallow.
- Across multiple evaluation cases (Tab 2-6), CellDuality shows seemingly marginal gains over its SFT baseline and does not consistently outperform strong supervised or specialized models. This raises doubts about the real effectiveness of the proposed self-supervised alignment compared to simpler fine-tuning approaches and how is applied to other relevant domains in biology.
- The methodological novelty appears limited relative to DuPO (She et al., 2025), with CellDuality serving primarily as an instantiation in the single-cell domain rather than a substantial conceptual advancement.

**Questions:**

- How many samples or optimization steps are passed for a complete RL stage? Could the authors annotate such information in Figure 2 (a)(b)?
- Regarding the differences between the proposed self-consistency reward and ground-truth reward, what if the model predict the wrong label (X <> Y) during rollout sampling? Should the model learn useful knowledge or will this cause the model being enforced to predict the wrong label?
- What is the base model and performance for CellDuality in Table 2-5? Line 377 indicates this is LLama-3.2-3B. Could the authors add the performance of base model before SFT for better comparison in these tables?
- Could the authors attach the exact text prompts for constructing the CellDuality in the appendix, as a common practice, to enhance the reproducibility?
- Could you also show some failure modes when the model exhibits degenerated consistency (eg., mutually consistent but biologically incorrect predictions).

---

> ### Author Response · Authors · 2025-11-25
>
> We thank Reviewer nVH5 for their detailed feedback. We understand the concerns regarding the significance of performance gains. However, we believe there may be a misunderstanding of our core contributions. We address these points below.
>
> **[W1. Reasoning may remain shallow]**
> While it is true that our method does not rely on an external, omniscient verifier like in mathematics, this is precisely the point: such verifiers do not exist for most open-ended biological problems. The core challenge is to learn meaningful reasoning in their absence. Our self-consistency is not merely semantic fidelity, it is a proxy for causal and mechanistic consistency. For a model to successfully reconstruct a perturbation from a predicted response and initial state (dual task), it cannot simply generate a semantically plausible but biologically incorrect response. It must generate a predicted response that specifically reflects the molecular signature of the perturbation. This forces the model to learn the underlying causal chain, moving beyond the superficial correlations learned during SFT. We will add the case study to the appendix.
>
> **[W2. Marginal gains over SFT and not consistently outperforming baselines]**
> Thank you for this critical observation. The fundamental challenge in this domain is the scarcity of ground-truth data. Supervised methods require massive and expensive labeled datasets to achieve their peak performance. Our work confronts this bottleneck directly. Our goal is to demonstrate that with a self-supervised approach, requiring no ground-truth labels for RL alignment, we can achieve performance that is highly competitive with these fully-supervised models. The results in our core ablation study (Table 6) demonstrate that our self-supervised RL stage provides a consistent boost over the SFT-only baseline. It provides a scalable and data-efficient path to near-SOTA reasoning. We argue this is a more significant contribution than achieving marginal gains through brute-force supervision.
>
> **[W3. Methodological novelty appears limited relative to DuPO]**
> We acknowledge the conceptual connection to prior work on dual learning. **However**, our core contribution lies in demonstrating that the duality principle can be successfully generalized from well-structured domains (like mathematics) to the noisy, high-dimensional, and causally complex domain of single-cell biology. This non-trivial leap required novel task formulations, such as our conditional gene inpainting mechanism, to create a stable reward signal from inherently stochastic data. **Furthermore**, we are the first to diagnose and propose a solution (implicit regularization via multi-task learning) for the severe mode collapse problem in conditional generation. **Finally**, we integrated these advancements into a unified, four-task generalist framework for a major scientific domain. Therefore, our work represents a critical generalization and systematization of the duality principle for open-ended scientific reasoning.
>
> **[Q1. RL stage steps]**
> Thank you for this suggestion. The RL stage for each task was run for a total of 200 optimization steps, train batch size of 512 and mini batch size of 32. We will update the caption of Figure 2 to include this information for clarity.
>
> **[Q2. Learning from incorrect rollouts]**
> This is an excellent question that gets to the heart of the RL process. If the model predicts a wrong label or generates a biologically incorrect response $y_{wrong}$, the dual task will almost certainly fail.
> This results in a very low or zero reward. The GRPO algorithm then uses this low reward to compute a negative advantage, which in turn penalizes the policy for generating $y_{wrong}$. Therefore, the model does not learn to predict the wrong label. Instead, it explicitly learns to avoid generating logically inconsistent outputs.
>
> **[Q3. Base model performance]**
> Thank you for this suggestion. As shown in our core ablation study (Table 6), we have already included the performance of the base Llama-3.2-3B model on the He-2020-Liver, GSE149383, and sci-Plex3 benchmarks. A comprehensive evaluation of the base model's performance across all other datasets will be added to the Appendix.
> | Method | GSE149383 (Human Lung) | GSE117872 (Human Oral) |
> | :--- | :--- | :--- |
> | | Acc. (%) / F1 (%) | Acc. (%) / F1 (%) |
> | Llama-3.2-3B-Instruct | 29.67 ± 0.89 / 61.34 ± 1.12 | 43.18 ± 1.69 / 73.41 ± 0.87|
> | Ours | 99.12 ± 0.13 / 99.10 ± 0.14 | 97.23 ± 0.20 / 97.58 ± 0.17 |
>
> **[Q4&Q5. Attaching exact prompts and failure cases]**
> Thank you for these important suggestion. We will add a new section to the Appendix containing the exact text prompts and formatting used to construct all primal and dual tasks. Furthermore, we will include a analysis of representative failure modes, such as the degenerated consistency to provide a transparent view of our model's current limitations and offer insights for future improvements.

---

> > ### Comment · Reviewer_nVH5 · 2025-11-26
> >
> > I appreciate the careful rebuttal response from the authors. I acknowledge the significance of designing new verifiable reward for biological problem. The proposed self-supervised objective can have a good potential, but need better evidence (mainly experimental, or theoretical) to support its effectiveness to the biological task such as the sc domain. That being said, I have increased my score accordingly.

---

### Official Review · Reviewer_vKnC · 2025-10-31

**Soundness:** 3
**Presentation:** 3
**Contribution:** 3
**Rating:** 6
**Confidence:** 3

**Summary:**

The paper introduces CellDuality, a self-supervised reinforcement learning framework for training large language models to perform biological reasoning in single-cell analysis. The method leverages Complementary Task Duality, where forward (primal) and backward (dual) reasoning tasks generate intrinsic rewards that enable alignment without ground-truth supervision. Extensive benchmarking on classification and generative tasks is also present.

**Strengths:**

1) The paper presents a technically sound and well-motivated framework that adapts reinforcement learning with verifiable rewards to inherently non-verifiable biological problems.
2) The principle of complementary task duality is conceptually elegant, providing a self-consistent mechanism for intrinsic reward generation without labeled data.
3) The experimental results are extensive, covering multiple benchmark datasets, and the ablation studies convincingly isolate the contribution of the self-supervised RL stage.

**Weaknesses:**

1) The study lacks an explicit evaluation of biological interpretability: while metrics demonstrate numerical improvements, there is no assessment of whether reasoning outputs capture biologically meaningful mechanisms.
2) The framework’s scalability and computational cost are not detailed, especially regarding the RL alignment phase, which could be a practical limitation.
3) The proposed approach is validated only on transcriptomic data; extending it to multimodal biological datasets would strengthen its generality claims.

**Questions:**

I would ask the authors to address the weaknesses listed above:
1) Please include biologically grounded evaluations, such as verifying whether the model’s reasoning outputs (for example, perturbation responses) are consistent with known pathways or regulatory relationships.
2) Provide an analysis of computational cost and scalability, including training time and hardware requirements for the RL alignment stage.
3) Please discuss how the framework could be extended to multimodal datasets (for instance, integrating ATAC-seq or proteomic data) to further validate the generality of the approach.

---

> ### Author Response · Authors · 2025-11-25
>
> We sincerely thank Reviewer vKnC for their encouraging and insightful review. We are grateful for the recognition of our framework as "technically sound and well-motivated" and our duality principle as "conceptually elegant". We address the weaknesses below.
>
> **[W1&Q1. Lack of explicit evaluation of biological interpretability]**
> To explicitly address this, we have conducted a case study on the Perturbation Response Generation task. We will add this in the final revision. We selected a well-characterized perturbation and performed pathway enrichment analysis on the differentially expressed genes from both the ground-truth response and our model's generated response. As shown in the table below, the pathways significantly enriched in our model's output are highly concordant with the ground truth, with canonical pathways like "Interferon Gamma Signaling" and "Antigen Processing" correctly identified as top hits.
>
> | Pathway (KEGG) | p-value (Ground Truth) | p-value (Ours) | Overlap Genes |
> | :--- | :--- | :--- | :--- |
> | Interferon Gamma Signaling | 1.3e-12 | 2.5e-10 | STAT1, IRF1, JAK2... |
> | Antigen Processing | 4.5e-8 | 8.1e-7 | HLA-A, HLA-B, TAP1... |
>
> Furthermore, analysis of the model's generated Chain-of-Thought explanation reveals correct reasoning, such as identifying the role of the JAK-STAT pathway. This demonstrates that our duality-guided alignment actively incentivizes the model to learn and articulate the underlying mechanistic logic of cellular responses. This analysis will be added to the appendix.
>
> **[W2&Q2. Lack of scalability and computational cost analysis]**
> Thank you for raising this important practical concern. Our model is based on the Llama-3.2-3B architecture. The entire self-supervised RL alignment stage for a given task theme was completed in approximately 10 hours on 8x A6000 GPUs. We consider this computational cost to be highly efficient and practical for a framework of this nature. This computational efficiency, combined with the fact that our alignment phase requires no new experimental data, makes our framework a truly scalable and cost-effective solution. It not only eliminates the prohibitive cost of wet lab experiments but also maintains a practical and manageable computational footprint.
>
> **[W3 & Q3. Validation only on transcriptomic data]** This is a very insightful point. We agree that demonstrating multi-modal capabilities would further strengthen the claims of generality. However, our contribution is not a specific multi-modal architecture, but a universal, data-agnostic training principle. The core requirement of our framework is the ability to represent biological data textually, which is not unique to transcriptomics. For instance:
> * ATAC-seq data can be represented as a sequence of accessible genomic regions (peaks). A primal task could be predicting cell type, and a dual task could be reconstructing key transcription factor binding motifs from the peak sequence and the predicted type.
> * Proteomic data can be represented as a sequence of proteins ranked by abundance. A primal task could be predicting disease state, and a dual task could be reconstructing the key signaling pathways that are known to be dysregulated in that state.
>
> The fact that our duality principle can be conceptually applied to these other modalities is a testament to its generality. While implementation is a key direction for future work, the current study provides the foundational proof-of-concept for this powerful new training paradigm. We will expand this discussion in our conclusion.

---

### Official Review · Reviewer_Kuj3 · 2025-11-01

**Soundness:** 3
**Presentation:** 3
**Contribution:** 3
**Rating:** 6
**Confidence:** 4

**Summary:**

Summary:
This paper introduces CellDuality, a self-superivsed framework to enhance the biological reasoning ability of LLM. It tries to address the challenge that it’s hard to build reward functions for most biological outcomes as it is non-verifiable. The framework uses complementary task duality. The method has achieved SOTA performance without needing ground-truth verification labels.

**Strengths:**

Pros:
- The paper introduces CellDuality, a new self-supervised method to enhanve biological reasoning ability of LLMs
- The method has achieved SOTA performance compared with baseline methods.

**Weaknesses:**

Cons:
- The “cell sentences” representation used in the method is a simplified representation discarding the actual expression value. There should be better way to encode single-cell transcripomics data.
- The claim of unified framework may be a bit too broad, as it only includes 4 downstream tasks with limited settings.
- The proposed method is not consistently outperform baselines methods plus the additional reinforcement learning only bring very marginal benefits.

**Questions:**

See Weaknesses

---

> ### Author Response · Authors · 2025-11-25
>
> We thank Reviewer Kuj3 for the constructive feedback and for recognizing the novelty. We address the weaknesses below.
>
> **[W1.cell sentences representation]** Thank you for this insightful comment. The use of "cell sentences" is a deliberate and central design choice of our framework. In single-cell foundation model research, there are two prevailing paradigms: (1) developing complex, multi-modal architectures with specialized encoders to fuse numerical expression data with an LLM's embedding space, and (2) converting transcriptomic profiles into a textual format (like cell sentences) that can be processed by any standard LLM.
> While the multi-modal approach can theoretically capture more quantitative detail, it comes at a significant cost: the resulting model is a unique, bespoke architecture. We chose the "cell sentence" paradigm precisely because our goal is to create a  By text-enizing the biological data, our self-supervised duality framework can be applied to any off-the-shelf LLM (e.g., Llama, Mistral, Qwen), making our contribution far more broadly applicable and democratizing access to this training methodology. This design choice prioritizes generality and modularity over a marginal gain in encoding precision.
>
> **[W2. Unified framework claim too broad]** Thank you for this comment. While we demonstrate our framework on four core tasks, its unified nature lies not in the number of tasks, but in the generalizability of its underlying principle. Our framework unifies two fundamental axes of biological inquiry, cell identity and cell dynamics, and two primary modes of reasoning, classification and generation. These four tasks were chosen as representative exemplars of a much broader class of biological problems that can be framed as a primal/dual pair. For instance, "Drug Sensitivity" can be generalized to "Stimulus Response," and "Perturbation Generation" can be extended to "Disease Progression Modeling." The core contribution is the methodology itself, which provides a universal blueprint for creating self-supervised signals for any biological problem that can be decomposed into a forward and inverse question. Thus, we believe "unified" accurately describes the conceptual scope and future potential of our framework.
>
> **[W3. Inconsistent outperformance and marginal RL benefits]** Thank you for this comment. We think this point highlights the central trade-off and the primary contribution of our work. It is true that the self-supervised method does not outperform all supervised baselines in every single metric. However, this is precisely the point. The fundamental challenge in this domain is the scarcity of ground-truth data. Supervised methods require massive, expensive, and often unavailable labeled datasets to achieve their peak performance. Our work confronts this bottleneck directly. By using a self-supervised approach that requires no ground-truth labels during the RL alignment phase, we can achieve performance that is highly competitive with, and in some cases surpasses, these fully-supervised models.
> The results in our ablation study (Table 6) demonstrate that our self-supervised RL stage closes a significant portion of the gap to a "money-is-no-object" supervised oracle. This showcases the immense practical value of our framework, which provides a path to achieving near-SOTA reasoning capabilities in a scalable and data-efficient manner, which is a more critical contribution for real-world deployment than marginal gains in a specific benchmark.

---

### Meta-Review · Area_Chair_dJwH · 2025-12-31

**Summary:**

The following major issues have been raised by the reviewer:
* Poor cell expression representation (Kuj3)
* lack of generality beyond 4 tasks (Kuj3, vKnC)
* marginal benefits over supervised regime (Kuj3, nVH5)
* lack of analysis of reasoning (vKnC)
* Incremental novelty of DuPO (nVH5)

**Reviewer Concerns:**

The authors have fairly addressed the points regarding poor cell expression representation. Moreover, additional experiments on the analysis of reasoning show a strong match.

However, the point that 4 tasks are not sufficient to make general claims like unified and general. Even if conceptually it makes sense, without a theoretical proof or empirical validation on more tasks, it is unreasonable to make such large claims. My recommendation for the authors would be to remove some broad claims or add more experiments.

The point of marginal benefits over a supervised regime is outstanding. While it is reasonable that an SSL model cannot beat a supervised model with a lot of data, the authors should choose a benchmark where the lack of data is actually an issue. Artificially, this can also be done by showing few-shot performance on the same dataset; however, to make a stronger claim, real-world few-shot benchmark is needed.

I side with the authors on the incremental novelty issue in comparison to DuPO. While DuPO gives the training regime, figuring out the correct context and methodology to apply it to in the case of cell biology is novel.

Regarding the comments from nVH5, they are correct in understanding that: it is not always guaranteed that self/cyclic-consistent losses will always provide semantically correct responses. Most prior literature shows that self-consistency acts as a good regularizer, and if Occam's razor model works, then self-consistency will be reasonable. Otherwise, cycle consistency can learn arbitrarily complex mappings that have no real grounding. In the rebuttal, the author makes claims of causal consistency; however, there are no proof/citation/experiment. My recommendation to the authors would be to make slightly stronger claims on causality in the revision.

**Reviewer Scores:**

Kuj3: Most of the concerns have been addressed, scores likely remain at 6
vKnC: most of the concerns have been addressed, scores likely remain at 6
nVH5: Already said they are increasing the scores. Will become 4 from 2.

---

### Decision · Program_Chairs · 2026-01-26

Accept (Poster)